# Changes in Learning Outcomes after Dietary Intervention in Preschoolers: A Pilot Study

**DOI:** 10.3390/nu13061797

**Published:** 2021-05-25

**Authors:** Faten Hasan, Jamie Jirout, Sarah Garzione, Sibylle Kranz

**Affiliations:** 1Department of Kinesiology, School of Education and Human Development, University of Virginia, Charlottesville, VA 22904, USA; fh4ua@virginia.edu; 2Department of Education, Development, Leadership and Foundations, School of Education and Human Development, University of Virginia, Charlottesville, VA 22904, USA; jj3wc@virginia.edu; 3College of Arts and Sciences, Formerly University of Virginia, Charlottesville, VA 22904, USA; sg2ja@virginia.edu; 4Department of Public Health Sciences, School of Medicine, University of Virginia, Charlottesville, VA 22904, USA

**Keywords:** preschooler, high-satiety diet, executive function, hunger

## Abstract

The executive functioning skill set, which includes working memory, cognitive flexibility, and inhibitory control, begins developing in early life and continues into adulthood. Preschoolers’ abilities to perform those skills may be influenced by diet. The purpose of this study was to explore the acute effects of consuming a low-GI diet compared to the usual childcare diet on preschoolers’ self-reported feelings of hunger and fullness and their performance on learning-associated tasks. This study was a prospective feeding trial in *n* = 20 children 3–4 years of age, completed in a laboratory setting where children attended “day camps” and consumed two days of usual diet (CON) and two days of low-GI (INT) diet. Learning outcomes were evaluated using select learning assessments including the Kansas Reflection-Impulsivity Scale for Preschoolers (KRISP), Track-it, Peg Tapping, and Happy/Sad. Repeated measures, full-factorial analysis of covariance revealed that diet was significantly related to impulsivity (*p* > 0.05), and univariate analysis of variance indicated that feelings of hunger and fullness differentially affected cognitive constructs in that feeling full improved impulsivity and attention, while feeling hungry improved inhibitory control. These findings highlight that the connection between diet and learning-related skills of children are independently mediated by both diet composition and feelings of hunger and fullness.

## 1. Introduction

Successful integration into the school system depends on a child’s learning-related skills such as executive function and self-regulation [1]. The executive functioning (EF) skill set, which includes working memory, cognitive flexibility, and inhibitory control, begins developing in early life and continues into adulthood [2]. The ability to maintain alert selective attention in the presence of endogenous and exogenous distractors indicates a child’s ability to succeed in a group classroom setting [3]. Controlling impulsivity is the ability to assess visual stimuli before reaction, which is helpful in problem solving and critical evaluation skills needed in the classroom setting [4]. These skills (EF, self-regulation, alertness, selective attention, and impulsivity) are influenced by neurological development and by long- and short-term environmental influences such as diet composition, which largely affects feelings of satiety [4].

One well-known factor contributing to satiety is the glycemic index (GI) of food, defined as the post-absorptive change in blood glucose compared to the standard of 50 g of glucose, or a slice of white bread, during the 2-h postprandial phase [5]. While GI is an absolute number, glycemic load (GL) reflects the actual metabolic response to eating a food (or meal) and is calculated by multiplying the grams of the food consumed with the GI of the food [6]. Therefore, GL can be understood as the ‘glucose removal burden’ following food intake. One effective technique to decrease the GL of meals is to substitute low-fiber foods with high-fiber foods [7,8,9]. Dietary fiber (soluble and insoluble fibers) slows glucose digestion and absorption, thus attenuating the postprandial rise and subsequent steep drop in blood glucose [10,11], which is linked with reduced release of orexigenic signals that likely lead to feelings of hunger [6,12,13,14,15]. The purpose of this study was to explore the acute effects of consuming a low-GI diet compared to the usual childcare diet on preschoolers’ self-reported feelings of hunger and fullness and their performance on a number of learning-associated tasks. We hypothesized that both feelings of hunger and feeling overly full would negatively affect a child’s performance on the learning-associated tasks.

## 2. Materials and Methods

This study was a prospective, randomly controlled clinical trial (registered with clinicaltrials.gov #NCT03861208) in *n* = 25 children 3–4 years of age (Table 1). Recruitment of children and their parents was conducted using targeted social media (Facebook) and fliers in nearby grocery stores and childcare centers. Inclusion criteria were local children aged 3–4 years old, and exclusion criteria were having food allergies, digestive diseases, or taking medications that affected food intake. Approval for the study was obtained from the local Institutional Review Board for Social and Behavioral Studies. All parents provided written informed consent, and participating children were asked for verbal consent before each study procedure. Data collection procedures were standardized through extensive training with research staff and followed a written protocol and scripts to ensure age-appropriate and consistent communication with children.

### 2.1. Study Design

Enrolled children were grouped into small cohorts based on their schedules and availability. A total of six cohorts attended the ‘day camps’ in groups of 3–6 children. Each cohort was randomly assigned (coin toss) to the order of two low-GI (intervention, INT) and two high-GI (control, CON) diets. The CON diet was based on usual meals and snacks at the local childcare centers; the INT diet was developed by using the same meals but substituting the ingredients with high-fiber ingredients, i.e., high-fiber versus refined grain pasta.

Upon arrival to the Diet and Nutrition (DAN) laboratory in the morning, parents completed a questionnaire to obtain demographic characteristics and provided a log of all foods and beverages the child consumed overnight and prior to arriving to the laboratory in the morning. Standing height and weight were measured using a digital height rod and body weight scale (Seca GmbH & Co, Hamburg, Germany) without shoes, to the nearest 0.1 cm and 0.1 kg, respectively. Weight status was calculated using the Centers for Disease Control and Prevention (CDC) BMI-for-age growth charts as per CDC guidelines [16].

### 2.2. Dietary Intervention

Two versions of usually consumed childcare menus were developed. The CON diet reflected typical foods served at local childcare centers, and the INT diet was designed to match the CON diet but using high-fiber and high-protein ingredients to result in a high-quality, low-GI meal plan. The foods of the INT diet were field-tested for acceptance in non-participants prior to the study. During the study, children were served the INT and CON diet for two non-consecutive days at “day camp” (seven hours (0830 to 1530)). Morning snack, lunch, and afternoon snack were served at 900, 1130, and 1445, respectively. The nutritional values, GI, and GL of the meals and snacks are reflected in Table 2.

All foods and beverages consumed (except water) were recorded by research staff using the plate waste method using calibrated research-grade food scales (Mettler Toledo, Columbus, OH, USA). In short, before each eating occasion, the children were asked to sit at a table, and the food was served on serving trays, specifically labeled for each participant. Children then consumed the foods ad libitum during a timeframe of approximately 15 min for each eating occasion. Initial serving sizes were based on typical, age-specific portions, and additional servings were weighed and recorded before being served onto the same tray. After the children stated they were finished eating, the trays were removed, and food waste was separated, weighed, and recorded. Combined foods such as spaghetti noodles and sauce were served separately, but children were allowed to combine them during the meal. The waste was separated as much as possible and weighed separately to determine the amount of each meal component consumed. Food intake was entered into Nutrient Data System for Research (NDSR, software version 2018, University of Minnesota, Minneapolis, MN, USA) and analyzed for total energy intake, food groups, macro-and micro-nutrients, GI, and GL.

### 2.3. Hunger and Fullness Measurement

Feelings of hunger and fullness were assessed pre-prandial as well as 60- and 120-min post-prandial the lunch meal using a validated, two-level comparison tool specifically developed for preschoolers, who usually have not yet developed serration and ordering skills [17]. The methodology is explained in detail elsewhere [17]. In short, training for the effective use of the tool was completed on the first study day. Trained research staff read a story from a picture book to the preschoolers, explaining the differences between having an “empty tummy” compared with having a “full tummy” and feeling “very” or “little” full, to explain the terminology. This was followed by a series of application practice examples consisting of telling short picture stories, after which children were asked to call out whether the fictitious characters had full or empty “tummies”. The children could respond verbally or point to silhouettes of “empty” or “full” tummies on a laminated answer sheet. The story and accompanying examples were repeated at the start of each study day. Self-reported feelings of hunger and fullness were based on sequential questions to first distinguish if the child felt hungry or full, followed by a second question to determine the relative level of hunger or fullness (“a little” or “very”). The combination of these responses was used to create a 4-point score, ranging from very hungry (1) to very full (4) [17]. For this study, ratings of the hunger questionnaire (HQ) were stratified into categories of “hungry” (1), “neutral” (2–3), and “full” (4) for the analyses of children’s hunger ratings 60-min postprandial the lunch meal.

### 2.4. Cognitive Measures

The cognitive assessments used in this study included learning-related measures of executive function and cognitive self-regulation [18,19], which included tasks relying on impulsivity and attentiveness (the Kansas Reflection-Impulsivity Scale for Preschoolers, KRISP [20]; Track-it [21]) as well as memory and inhibitory control (Peg Tapping [18]; Happy/Sad [22,23]). Each task is described in more detail below [2]. Each member of the research staff was extensively trained to administer each specific learning assessment, and a script was used for standardization.

All cognitive tasks were administered to each child each day of the study at 1145 and 1245 to correspond with the collection of data from the HQ. The KRISP and Happy-Sad tasks were administered immediately after lunch, and the Peg-Tapping and Track-it were administered one hour after lunch. This was done to assess immediate and delayed responses to the dietary intervention. Each participant completed tasks one at a time with a researcher, while the remaining children engaged in free play. Participants were taken to a room separate from the play area during tasks so as to omit distraction. Once a participant completed a given task, he or she returned to free play with the other children.

#### 2.4.1. Kansas Reflection-Impulsivity Scale for Preschoolers (KRISP)

The KRISP task assesses impulsivity by measuring a child’s ability to assess visual stimuli before reacting to them. This test is frequently used to evaluate a child’s tendencies to scan and assess multiple stimuli and avoid impulsive reactions, a valuable skill in classroom learning [4]. For the test assessment, the researcher held a binder with laminated pages of images. On a given spread of two pages, opened vertically, the top page had an image of a single recognizable object (i.e., cat, mouse, pail, etc.), and the bottom page contained between four and eight similar drawings, one of which was the exact same as the top image, and the rest with slight variations. Variations included changes in shading patterns, lack of certain features (such as a cat without a tail), or other similar adjustments from the original drawing.

At the beginning of the task, the researcher instructed the child to point to the image on the bottom page that was the exact same as the top page. If the child indicated the correct image, the researcher turned to the next set of images; however, if they indicated the incorrect image, the researcher repeated the same instructions. A second incorrect response resulted in repetition of the same instructions again. After three incorrect responses, the researcher moved onto the next page. This process was repeated for 15 additional trials of different images. Each set of images was scored according to the number of incorrect responses on that page (0 = first response correct, 1 = one incorrect response, 2 = two incorrect, 3 = three incorrect), and the final score consisted of a total count of incorrect responses in the 16 total trials.

#### 2.4.2. Track-It

Track-it assessed how endogenous and exogenous factors manipulated selective sustained attention on a task that a young child would likely not complete without an adult’s request. Exogenous factors include external stimuli, while endogenous factors include intentional control on the point of attention [3]. While infants and toddlers are prone to distraction from exogenous factors, as a child develops, endogenous factors play an increasing role in attentiveness. By measuring a child’s errors in Track-it, one can assess the ability of a child to remain alert, assess sensory input, and sustain attention in the presence of distraction stimuli [21].

The Track-it test was administered on a laptop program. On the screen was a 3 × 3 grid of nine squares, six of which contained distraction stimuli, or colorful shapes (i.e., yellow heart, grey cross). The grid boxes were colored so as to help the participant to distinguish between them. Before beginning the trials, an additional object, such as a shape, was indicated as the target by a red circle.

While viewing the initial screen, the child was informed that the target would move around the 3 × 3 grid and then disappear. The child was asked to visually follow the moving target and indicate at which square the target disappeared. The experiment consisted of one practice round and two trial rounds, scored for indication of the correct target square (1 = correct, 0 = incorrect). The individual trial scores were added together for the final score. Though the Track-it task sometimes has a second section assessing participant memory, this section was discounted due to high levels of misunderstanding and error among the children.

#### 2.4.3. Happy-Sad

The Happy-Sad task measures inhibitory control, cognitive regulation, and executive function. It was developed as a version of the Stroop Color–Word Test for non-literate groups and serves to assess ability to follow instruction in the presence of distracting stimuli [24]. This requires children to keep specific rules in memory while resisting the urge to verbalize their previously taught semantic associations with these images, which is particularly difficult in this task due to the deep associations with emotions from infancy [22].

When giving the Happy-Sad task, the researcher held a pile of square flashcards with smiling or frowning faces. Other than the smile or frown, all of the faces were exactly the same. When explaining the task, the researcher held up the flashcards one at a time and asked the child to say “happy” when they saw a frowning face or “sad” when they saw a smiling face. Regardless of a correct or incorrect response, the researcher moved on to the next flashcard.

The researcher administered four practice rounds wherein the child was praised for any correct response. If the child answered incorrectly and did not understand the task, the researcher repeated the same instructions. Once the task was understood, the researcher proceeded with twenty trials (discounting the practice rounds) and did not praise or reprimand the child for correct or incorrect responses. The researcher counted every time the child vocalized the correct response (1 = correct, 0 = incorrect). Scores consisted of the total number of correct responses.

#### 2.4.4. Peg Tapping

The Peg Tapping task measures working memory and inhibitory control, which follow frontal lobe development and are associated with greater future academic achievement [25]. Development of these executive function skills helps a child to problem solve in a goal-oriented manner [2].

During the Peg Tapping task, the researcher and child each held a pencil, and the following explanation of the task was given: When the researcher tapped the back of their pencil once on the table, the child was to tap the table two times with their pencil. Similarly, when the researcher tapped twice, the child was to follow with one tap.

The researcher then initiated two rounds of practice. For these two practice rounds, the child was praised for any correct responses. If both practice trials were incorrect, these rounds were discounted, and instructions and practice rounds were repeated. Once the child understood the task, the researcher then continued through 16 trials (including the final 2 practice trials) without any praise or reprimand for responses. Children had to keep the rules of when to tap and how many times in memory while inhibiting the impulse to mimic the researcher’s tapping or tap extra times. The researcher counted every time that the participant tapped the correct number of times on the table (1 = correct response, 0 = incorrect response). Final scores consisted of the total number of correct responses in the 16 trials.

### 2.5. Statistical Analysis

Statistical analyses were conducted using IBM SPSS Statistics for Macintosh, Version 27.0 [26]. Three children did not participate in at least two of the four required study days and were excluded from the analysis. Two additional children (ages 36 and 37 months) did not understand the learning tasks and were also excluded from the analysis, resulting in a final sample size of *n* = 20. All data are presented as mean ± SD unless otherwise stated. Paired t-tests (two-tailed) were used to compare energy and nutrient intake and mean GI and GL between the CON and INT diets. Proportional odds ratios were used to investigate differences in hunger and fullness ratings between the CON and INT diets. Repeated measures, full-factorial analysis of covariance (ANCOVA) was used to test for differences between the study conditions (INT, CON) and the day of the study (day 1, day 2) on each of the learning assessments, using both condition and day as within-subjects variables and controlling for age and sex. Univariate analysis of variance (ANOVA) was used to test for associations between feelings of hunger and performance on each of the learning assessments on each day of the study, using hunger rating and sex as factors and controlling for age. Statistical significance was accepted as *p* ≤ 0.05.

## 3. Results

Feelings of hunger and fullness were similar on both diets on study day 1 (*p* = 0.13) and day 2 (*p* = 0.38), based on the odds ratio of reporting “hunger”. For instance, across both days, 79% and 67% of the children reported feeling “hungry” prior to lunch, and 53% and 56% of the children reported feeling “full” after lunch on the CON and INT diets, respectively.

Mean performances on learning-associated tasks are presented in Table 3. We only report significant analyses here and provide all p-values and partial eta-square values comparing differences in learning task performance by condition in Table 4, with estimated means and standard error graphed by condition and day in Figure 1. For the KRISP task, we observed significant main effects for age and day. As expected, children’s performance generally improved with age, with scores decreasing (improving) as age increased (*r* = −0.65, *p* = 0.002). Children scored significantly better (lower score) on the second day than the first, though this was qualified by three interactions. There were interactions between condition and day (F(1,12) = 5.18, *p* = 0.04), sex and day (F(1,12) = 4.56, *p* = 0.05), and a three-way interaction among study day, condition, and age (F(1,12) = 4.77, *p* = 0.05). For females, scores were significantly worse (higher) on day 1 (M = 10.42, SE = 1.53) than on day 2 (M = 5.34, SE = 2.36). For males, scores were not different between day 1 (M = 7.85, SE = 0.911) and day 2 (M = 7.06, SE = 1.40). On day 2 of each diet, we observed a significant main effect of condition (*p* = 0.023), qualified by an interaction between condition and age. Controlling for age, children performed better on day 2 of the INT diet (M = 5.22, SE = 1.05) compared to day 2 of the CON diet (M = 6.94, SE = 1.20). However, performance on day 1 of each diet was similar (M = 9.38, SE = 1.17 and M = 8.13, SE = 0.95 on the CON and INT diets, respectively; *p* = 0.99).

There were no significant effects observed for performance on the Happy/Sad task. For the Peg Tapping task and Track-it task, we observed significant main effects only for age (*r* = 0.60, *p* = 0.005 and *r* = 0.78, *p* < 0.001, respectively).

All p-values and partial eta-square values comparing differences in learning task performance by hunger/fullness ratings on each study day are provided in Table 5, and significant effects are reported here. Overall, there were no global differences in performance on learning tasks between children feeling hungry, neutral, or full. For the KRISP task, we observed a significant effect of age on both days of the CON diet and day 1 on the INT diet. Children’s performance on the task improved as age increased (*r* = −0.58, *p* = 0.02; *r* = −0.54, *p* = 0.02; and *r* = −0.66, *p* = 0.002, respectively). Similarly, children’s performance improved as age increased on the Happy/Sad task on day 2 of the CON diet (*r* = 0.59, *p* = 0.009) and the Peg Tapping task on day 2 of the CON and INT diets (*r* = 0.63, *p* = 0.004 and *r* = 0.61, *p* = 0.005, respectively), and for the Track-it task on day 2 of the CON diet (*r* = 0.62, *p* = 0.02) and both days of the INT diet (*r* = 0.79, *p* < 0.001 and *r* = 0.82, *p* < 0.001, respectively). Finally, for the Track-it task, we observed a significant main effect of sex on day 2 of the CON diet in that females (M = 2.4, SE = 0.48) performed better than males (M = 0.88, SE = 0.32).

For the KRISP task, pairwise comparisons revealed that on day 1 of the CON diet, children who reported feeling full (M = 5.25, SE = 1.72) performed significantly better than those who reported feeling neutral (M = 12.18, SE = 1.54; *p* = 0.01). Conversely, for the Happy/Sad task, on day 2 of the CON diet, children who reported feeling hungry (M = 18.04, SE = 1.28) performed better than those who reported feeling neutral (M = 13.92, SE = 1.45; *p* = 0.05). Similarly, on day 1 of the INT diet, children who reported feeling hungry (M = 17.0, SE = 2.67) performed significantly better on the Happy/Sad task than those who reported feeling full (M = 10.10, SE = 1.84; *p* = 0.05). Finally, for the Track-it task, children who reported feeling full (M = 2.18, SE = 0.24) performed better than those who reported feeling neutral (M = 0.96, SE = 0.40).

## 4. Discussion

This study showed that preschoolers performed significantly better on impulsivity tasks while consuming the INT diet, but responses were not significantly different for inhibitory control, working memory, or attention tasks.

To our knowledge, this is the first dietary intervention study to investigate the effects of GI/GL on learning-related tasks in preschoolers. Several cross-sectional and cohort studies have been conducted to investigate the role of diet quality and food patterns on behavior, brain health and function, and academic performance in preschoolers. Feinstein et al. [27] found that in three-year old children, consuming “junk food” was negatively associated with level of school attainment, while consuming a “health conscious” dietary pattern was associated with greater school attainment [27]. Similarly, a study in mainland China found that preschoolers consuming a diet high in “processed” foods had greater odds for having symptoms associated with attention-deficit/hyperactivity disorder [28]. Northstone et al. [29] had similar findings, in that consumption of a more “processed” dietary pattern, consisting of high fat and sugar content, at three years old, was negatively associated with IQ at age 8.5 years [29]. A proposed explanation for these findings is that “processed” dietary patterns are high in sugar and refined carbohydrates, and thus likely a high-GI diet; however, the GI or GL of the children’s diets were not assessed. Therefore, these observational studies are in line with our finding that a low-GI, high-quality diet may positively impact children’s ability to learn by improving impulsivity.

In older children, research on the acute role of breakfast on mood, focus, and ability to learn in children and adolescents has shown that low-GI breakfasts beneficially affect mood (more confident, less sluggish, less hungry) [30] and performance on cognitive tasks (memory and ability to sustain attention) compared to a high-GI breakfast [30,31,32]. Conversely, Brindal et al. [33] found that low-GL versus medium- or high-GL breakfasts did not significantly alter cognitive function, self-reported satiety, or energy intake at subsequent lunch meals in 10–12 year old children; however, these diets also varied in macronutrient content and dairy food composition [33]. Studies using chronic manipulation of diet quality in schools have also yielded promising results on learning behavior. A 12-week intervention to promote healthier school food at lunchtime in 3rd to 5th grade children improved alertness and teacher–pupil on-task engagement [34]. A similar effort to improve school meal options in primary schools in the UK found that it significantly improved English and science educational outcomes [35]. Thus, it is evident that both diet quality and GI/GL manipulation likely affect both short-term and long-term learning behaviors such as focus and attention, as well as cognition. Two potential explanations for this outcome are the influences of diet quality and GI/GL on behavioral outcomes.

Changes in performance on cognitive tasks may, in part, be due to the influence of GI/GL on metabolic processes. Although this was not measured in our study, high-glycemic foods are rapidly digested and absorbed, thus causing a sharp postprandial rise and subsequent steep drop in blood glucose, which triggers feelings of hunger [10,11]. The brain preferentially utilizes blood glucose as its primary energy source; thus, the rapid fluctuations in glucose levels may impair glucose availability to brain tissue and potentially affect cognitive processes. Although this area is vastly understudied, it is reasonable to assume that this may disproportionally affect children due to the intensive brain development during childhood that dramatically increases glucose demands and the rate of glucose utilization, which can be as high as up to twice that of an adult [32]. Furthermore, chronic consumption of a high-glycemic diet and the associated glucose fluctuations have been linked to metabolic disturbances and increased risk of type 2 diabetes [36]. In 4–6-year-old children, metabolic markers such as fasting glucose and insulin as well as the Homeostatic Model of Assessment for Insulin Resistance (HOMA-IR) were all found to be inversely associated with performance on inhibitory control and may have partially explained the long-term effects of diet on cognitive performance [37].

Secondly, the role of feelings of hunger and fullness on learning-associated tasks were investigated in our study. While we found no effect of GI or GL on hunger and fullness ratings or overall energy intake, similarly to other studies in preschoolers [17,38] and older children [33], we did find that full children performed better on impulsivity and attention tasks, hungry children performed better on the inhibitory control task, and there was no effect of hunger/fullness on the working memory task. This suggests that certain brain functions may be differentially impacted by feelings of hunger and fullness, potentially explaining the equivocal findings of diet manipulation on cognition. However, it is well-established that omission of food intake, namely breakfast, negatively affects cognition (reaction time, working memory, and inhibitory control) and ability to learn (attention and visual–spatial memory) [39,40], especially in nutritionally at-risk children [41]. These findings are thought to be mediated by both physiological mechanisms (e.g., levels of neuronal glucose uptake, glucose mediated insulin delivery, acetylcholine synthesis, and carbohydrate-induced cortisol secretion) and subjective mechanisms (feelings of hunger mediated by release of orexigenic signals (e.g., elevated ghrelin) and changes in mood, alertness, and motivation) [6,12,13,14,15,42]. While this explains the improved performance on impulsivity and attention tasks in full children, the improved performance on inhibitory control tasks in hungry children was unexpected.

Limitations of this study include the short-term nature of our results and small sample size. We measured only the acute effect, within one day, of serving the diets, and it is known that metabolic adaptations require longer-term exposure to low-GI diets of at least two weeks; it is not known if the same adaptation time is needed in young children. It should be noted that up to 10% of children in the United States experience “faltered growth” or “failure to thrive” [43], largely due to malnutrition from inadequate macro- and micro-nutrient intake [44,45]. Failure to thrive can affect both growth and development, including neurocognitive development [45]. In these situations, a low-GI diet may inadvertently reduce food intake and exacerbate the poor outcome for the child [46]. In addition, the learning-associated tasks used were only reflective of processes that were previously found to facilitate learning, rather than measuring learning itself. Directly assessing the effects of diet on learning would require tightly controlled, long-term interventions, as numerous external factors contribute to learning and cognition in children. Further, the wide range in age of our study population led to a large variation in performance on the tasks. Although we did control for age in our analyses, future studies should utilize additional control assessments such as baseline cognitive ability, and preliminary testing should be used to ensure that the changes in task performance are relative to the child’s development.

## 5. Conclusions

In conclusion, this pilot study based on a dietary intervention showed that preschoolers could successfully lower GI/GL when offered a high-quality diet. We found indications that diet related to impulsivity. Additionally, we found that feelings of hunger and fullness differentially affected cognitive constructs in that feeling full improved impulsivity and attention, while feeling hungry improved inhibitory control. These findings highlight that the connection between diet and learning-related skills of children are independently mediated by both diet composition and feelings of hunger and fullness. Future studies in larger and more diverse samples and conducted over longer periods of time that also include additional measures on learning, cognition, and other important factors in children’s ability to learn are needed to better understand the relationship between diet and learning in preschool-age children.

## Figures and Tables

**Figure 1 nutrients-13-01797-f001:**
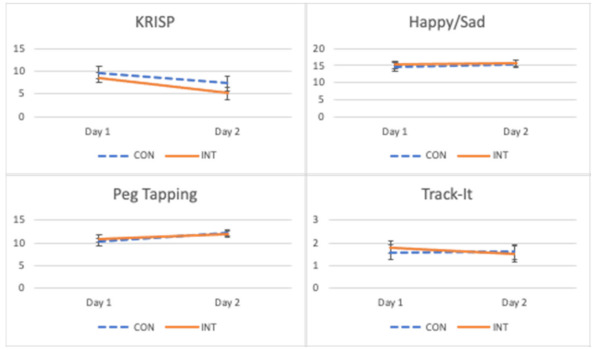
Estimated means and standard error graphed by condition (CON and INT) and day (1 and 2). CON: control diet; INT: intervention diet.

**Table 1 nutrients-13-01797-t001:** Characteristics of preschoolers in the sample.

		Total Sample (*n* = 20)
Age (months)	≤48	50
	>48	50
Sex	M	70
	F	30
BMI Percentile (%)	<5	5
	5–85%	65
	>85	30

Data are in %.

**Table 2 nutrients-13-01797-t002:** Dietary characteristics of preschoolers on each diet.

	CON	INT
Energy, kcal	500.0 ± 167.5	472.4 ± 236.2
Protein, g (%)	11.9 ± 5 (9%)	14.7 ± 8.7 (12%)
Carb, g (%)	87.9 ± 30.1 (69%)	85.1 ± 46.4 (68%)
Added Sugar (g)	10.3 ± 5.3	10.5 ± 7.9
Fiber, g	6.3 ± 2.6	14.8 ± 9.1 ^a^
Soluble fiber, g	1.9 ± 0.6	8.3 ± 5.5 ^a^
Insoluble fiber, g	4.5 ± 1.7	6.6 ± 4.1 ^a^
Fat, g (%)	12.5 ± 5.5 (22%)	11.0 ± 5.2 (20%)
Glycemic Index (glucose reference)	58.6 ± 5	46.6 ± 3.4 ^a^
Glycemic Index (bread reference)	83.8 ± 7.1	66.5 ± 4.8 ^a^
Glycemic Load (glucose reference)	47.7 ± 17.3	25.4 ± 17.4 ^a^
Glycemic Load (bread reference)	68.2 ± 24.7	36.3 ±24.9 ^a^

CON, control diet; INT, intervention diet. ^a^ Significantly different at *p* < 0.05.

**Table 3 nutrients-13-01797-t003:** Mean performance across both days on learning-associated tasks on CON and INT diets.

	CON	INT
Impulsivity	7.97 ± 5.70	6.64 ± 4.60
Inhibitory control	15.43 ± 4.16	14.82 ± 4.56
Working memory	11.41 ± 4.10	11.54 ± 3.71
Attention	1.48 ± 1.23	1.56 ± 1.27

CON: control diet; INT: intervention diet.

**Table 4 nutrients-13-01797-t004:** Repeated measures ANOVA: diet vs. learning task.

	Main Effects		Covariates		Interactions						
Measure Type	Cond	Day	Sex	Age	Cond × Day	Cond × Sex	Cond × Age	Day × Age	Day × Sex	Cond × Day × Sex	Con × Day × Age
Impulsivity	*p* = 0.12 η^2^ = 0.19	*p* = 0.03 * η^2^ = 0.04	*p* = 0.84 η^2^ = 0.003	*p* = 0.03 * η^2^ = 0.33	*p* = 0.04 * η^2^ = 0.30	*p* = 0.88 η^2^ = 0.002	*p* = 0.16 η^2^ = 0.16	*p* = 0.07 η^2^ = 0.24	*p* = 0.05 * η^2^ = 0.28	*p* = 0.94 η^2^ = 0.00	*p* = 0.05 * η^2^ = 0.28
Inhibitory control	*p* = 0.50 η^2^ = 0.04	*p* = 0.42 η^2^ = 0.06	*p* = 0.57 η^2^ = 0.03	*p* = 0.06 η^2^ = 0.27	*p* = 0.64 η^2^ = 0.02	*p* = 0.62 η^2^ = 0.02	*p* = 0.54 η^2^ = 0.03	*p* = 0.49 η^2^ = 0.04	*p* = 0.68 η^2^ = 0.02	*p* = 0.17 η^2^ = 0.15	*p* = 0.60 η^2^ = 0.02
Working memory	*p* = 0.19 η^2^ = 0.11	*p* = 0.21 η^2^ = 0.10	*p* = 0.27 η^2^ = 0.08	*p* = 0.001 * η^2^ = 0.53	*p* = 0.21 η^2^ = 0.10	*p* = 0.78 η^2^ = 0.006	*p* = 0.22 η^2^ = 0.10	*p* = 0.39 η^2^ = 0.05	*p* = 0.69 η^2^ = 0.01	*p* = 0.63 η^2^ = 0.02	*p* = 0.26 η^2^ = 0.09
Attention	*p* = 0.70 η^2^ = 0.02	*p* = 0.99 η^2^ = 0.00	*p* = 0.93 η^2^ = 0.001	*p* = 0.001 * η^2^ = 0.76	*p* = 0.61 η^2^ = 0.03	*p* = 0.67 η^2^ = 0.02	*p* = 0.66 η^2^ = 0.03	*p* = 0.96 η^2^ = 0.00	*p* = 0.69 η^2^ = 0.21	*p* = 0.50 η^2^ = 0.06	*p* = 0.67 η^2^ = 0.02

CON: control diet; INT: intervention diet; Day: 1 and 2. * Significant at *p* ≤ 0.05.

**Table 5 nutrients-13-01797-t005:** One-way ANOVA: hunger ratings vs. learning task.

	CON								INT							
	Day 1				Day 2				Day 1				Day 2			
Measure Type	Hunger	Sex	Age	Hunger × Sex	Hunger	Sex	Age	Hunger × Sex	Hunger	Sex	Age	Hunger × Sex	Hunger	Sex	Age	Hunger × Sex
Impulsivity	*p* = 0.06 η^2^ = 0.44	*p* = 0.90 η^2^ = 0.002	*p* = 0.02 * η^2^ = 0.44	*p* = 0.28 η^2^ = 0.12	*p* = 0.51 η^2^ = 0.11	*p* = 0.99 η^2^ = 0.00	*p* = 0.02 * η^2^ = 0.36	*p* = 0.40 η^2^ = 0.14	*p* = 0.20 η^2^ = 0.22	*p* = 0.53 η^2^ = 0.03	*p* = 0.05 * η^2^ = 0.27	*p* = 0.83 η^2^ = 0.03	*p* = 0.47 η^2^ = 0.11	*p* = 0.85 η^2^ = 0.003	*p* = 0.73 η^2^ = 0.01	*p* = 0.69 η^2^ = 0.01
Inhibitory control	*p* = 0.94 η^2^ = 0.01	*p* = 0.61 η^2^ = 0.03	*p* = 0.43 η^2^ = 0.06	*p* = 0.11 η^2^ = 0.23	*p* = 0.12 η^2^ = 0.30	*p* = 0.84 η^2^ = 0.004	*p* = 0.01 * η^2^ = 0.41	*p* = 0.46 η^2^ = 0.12	*p* = 0.08 η^2^ = 0.32	*p* = 0.55 η^2^ = 0.03	*p* = 0.70 η^2^ = 0.01	*p* = 0.54 η^2^ = 0.09	*p* = 0.56 η^2^ = 0.09	*p* = 0.98 η^2^ = 0.00	*p* = 0.08 η^2^ = 0.21	*p* = 0.65 η^2^ = 0.02
Working memory	*p* = 0.73 η^2^ = 0.05	*p* = 0.18 η^2^ = 0.14	*p* = 0.06 η^2^ = 0.25	*p* = 0.71 η^2^ = 0.05	*p* = 0.72 η^2^ = 0.05	*p* = 0.48 η^2^ = 0.04	*p* = 0.04 * η^2^ = 0.30	*p* = 0.94 η^2^ = 0.01	*p* = 0.25 η^2^ = 0.20	*p* = 0.39 η^2^ = 0.06	*p* = 0.80 η^2^ = 0.005	*p* = 0.40 η^2^ = 0.13	*p* = 0.09 η^2^ = 0.31	*p* = 0.04 * η^2^ = 0.29	*p* = 0.002 * η^2^ = 0.54	*p* = 0.46 η^2^ = 0.04
Attention	*p* = 0.46 η^2^ = 0.13	*p* = 0.52 η^2^ = 0.04	*p* = 0.07 η^2^ = 0.27	*p* = 0.80 η^2^ = 0.04	*p* = 0.22 η^2^ = 0.31	*p* = 0.03 * η^2^ = 0.46	*p* = 0.03 * η^2^ = 0.45	*p* = 0.44 η^2^ = 0.19	*p* = 0.07 η^2^ = 0.45	*p* = 0.44 η^2^ = 0.07	*p* = 0.001 * η^2^ = 0.74	*p* = 0.24 η^2^ = 0.28	*p* = 0.53 η^2^ = 0.12	*p* = 0.60 η^2^ = 0.03	*p* = 0.001 * η^2^ = 0.67	*p* = 0.97 η^2^ = 0.00

CON: control diet; INT: intervention diet; Day: 1 and 2. * Significant at *p* ≤ 0.05.

## Data Availability

This study is nested within a project registered with clinical trials.gov, and data will be uploaded upon completion of the study.

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
