# Peer review of "Changes in Learning Outcomes after Dietary Intervention in Preschoolers: A Pilot Study"

_nutrients, 2021, doi:10.3390/nu13061797_

Round 1
Reviewer 1 Report
This is an original study which aims was to explore the acute effect of consuming a low-GI diet compared to the usual childcare diet on preschoolers’ self-reported feelings of hunger and fullness and their performance on learning-associated tasks.
Overall, I believe this manuscript meets the quality standards for publication in Nutrients journal. It is easy to read, the introduction is good, uses an adequate methodology and results are adequate to respond to the objective. The manuscript provides scientific evidence on intake of low-GI diet compared to the usual childcare diet on preschoolers' self-reported feelings of hunger and fullness and their performance on learning-associated tasks.
Reviewer 2 Report
The authors, in a small clinical pilot study of young children, compared consumption of a low glycemic-index diet for two days with the children’s standard diet for 2 days in a laboratory setting and determined that feeling full improved impulsivity and attention while feeling hungry improved inhibitory control.
The abstract was informative. The background briefly summarized the observation that dietary fibers reduce the fluctuation in post-prandial blood glucose. The factors related to development of children’s executive function (EF) skills including self-regulation, alertness, selective attention, and impulsivity, which working memory; cognitive flexibility; and inhibitory control were reviewed. The authors tested the hypothesis that feelings of hunger and/or feeling overly full negatively affect the child’s performance on learning associated tasks.
The study was well-designed and the method section was detailed in providing a concise description of the experimental procedures, with adequate regulatory information.
The description of the statistical analysis was appropriate.
The results were thoroughly described. The authors described 4 variables that were used to evaluate the diet intervention that were assessed by KRISP, Happy/Sad, Peg Tapping and Track-It, which resulted in quantitative information supporting their claims.
The discussion and the conclusions were clear. It will be important to evaluate the long-term effect on growth and muscle development of a lower GI/GL diet, since appropriate energy consumption is necessary for young children’s growth and development.
This is a nice study and analysis of the lower GI/GL diet for consumption by young children, was well-done. This area of research deserves more emphasis. This type of research is an opportunity to advance preventive medicine and translate it into the clinic.
Suggestions:
Besides the short-term nature of the study results and small sample size, please discuss the effects of low energy consumption on growth and development of young children.
